# An Improved Backoff Scheme and Its Performance Analysis for Full Duplex MAC Protocols in VLC Networks [note 1]

**DOI:** 10.3390/s21248263

**Published:** 2021-12-10

**Authors:** Yuta Sawa, Kosuke Sanada, Hiroyuki Hatano, Kazuo Mori

**Affiliations:** Department of Electrical and Electronics Engineering, Mie University, 1577 Kurimamachiya-cho, Tsu-shi 514-8507, Japan; sawa@com.elec.mie-u.ac.jp (Y.S.); hatano@elec.mie-u.ac.jp (H.H.); kmori@elec.mie-u.ac.jp (K.M.)

**Keywords:** IEEE 802.15.7, visible light communication, full duplex, MAC protocol, Markov-chain model, theoretical analysis

## Abstract

IEEE 802.15.7 Visible Light Communication (VLC) networks suffer from performance degradation caused by the hidden device collisions due to the directional transmission with narrow beamwidth. One of the solutions for mitigating the hidden device collisions is to employ a full-duplex transmission technique. As a side effect of the full-duplex transmission in the VLC networks, however, the data-packet discard due to the retransmission limitation occurs frequently in the networks. This paper proposes an improved backoff scheme and its performance analysis to suppress the packet discard. The proposed backoff scheme increases the Backoff Exponent (BE) and the Number of Backoff stage (NB) in IEEE 802.15.7 only when the data packet transmission fails. To evaluate the system performance theoretically, this paper also provides the Markov-chain model for channel access with the proposed scheme. The performance evaluations through simulation and theoretical analysis show the effectiveness of the proposed scheme.

## 1. Introduction

Visible Light Communication (VLC) using Light-Emitting Diode (LED) has recently attracted attention. Because visible light has a higher frequency (430–770 THz) than radio frequency (up to 3 THz), it enables high rate communication, less interference caused by other electromagnetic waves [1]. IEEE 802.15.7 [2] has been standardized as a standard for VLC networks. This standard specifies the Physical (PHY) and Medium Access Control (MAC) layers for VLC networks, in which the Carrier Sense Multiple Access with Collision Avoidance (CSMA/CA) is employed. However, signal collision induced by hidden devices causes more serious performance degradation [3,4]. In radio networks with omni antenna, the transmitted signal propagates in all directions around the antenna. Therefore, the frame collisions induced by the hidden-device problem occur when the devices are deployed outside of the communication range of the transmitted signal. On the other hand, in VLC networks, the signal propagates in a limited direction due to the directional transmission with narrow beamwidth. Therefore, it is difficult for VLC-network devices to detect signals transmitted by the neighbors even if the distance between the devices is short [4]. For achieving the performance improvement of VLC networks, it is necessary to mitigate the effect of hidden devices.

One of the approaches for mitigating the hidden device collisions is to apply Full Duplex (FD) communication to VLC networks [5,6,7,8,9,10]. FD wireless communication can be easily applied to bi-directional communication in VLC networks compared with radiofrequency communication. This is because VLC devices use separate components for their transmission and reception circuits, such as LEDs and photodetectors. Therefore, the research regarding FD VLC networks have increasingly been reported in recent years, and some of this research has proposed MAC protocols for FD VLC networks [5,6,7,8,9,10]. In the FD VLC networks, a coordinator sends a busytone signal to all devices while receiving a data packet from its device. The busytone signal prevents hidden devices from transmitting their data packets, leading to significantly reducing collision induced by hidden devices. As its side effect, however, the backoff scheme specified in IEEE 802.15.7 based on FD VLC networks has caused serious data-packet discards even if the packets are not involved in any collisions. This is because the device may keep increasing its values of the Backoff Exponent (BE) and the Number of Backoff stage (NB) at every Clear Channel Assessment (CCA) due to the busytone signal transmitted from the coordinator.

This paper proposes an improved backoff scheme to reduce the serious packet discard and its performance evaluation based on the theoretical analysis. In the proposed backoff scheme, the BE and NB values in IEEE 802.15.7 are increased only when the data packet transmission fails due to packet collision. Namely, the BE and NB have kept the current values for detecting the channel busy at CCA. As a side effect of the applied modification of the proposed scheme, however, it would cause a larger transmission delay. Therefore, to suppress the larger transmission delay, a limitation for keeping the BE and NB is applied to the proposed scheme. This paper also provides the theoretical model with a Markov chain for channel access with the proposed scheme, which includes the limitation for keeping the BE and NB. Through the theoretical analysis and computer simulation, we show the effectiveness of the proposed scheme. In addition, we indicate an appropriate value for the limitation for keeping the BE and NB.

The rest of this paper is organized as follows: Section 2 provides an overview of related work, specifically IEEE 802.15.7 CSMA/CA and FD-RTS/CTS aided CSMA/CA scheme. Section 3 proposes the improved backoff scheme for the FD-RTS/CTS scheme in VLC networks. In Section 4, we present theoretical analysis to evaluate the performance of the proposed backoff scheme. Section 5 presents the effectiveness of the proposed backoff scheme through theoretical analysis and computer simulation. Finally, Section 6 concludes the paper.

## 2. Related Work

### 2.1. IEEE 802.15.7 CSMA/CA and Hidden Device Problem in VLC Networks

In the VLC networks, the multiple devices will access communication channels following the CSMA/CA algorithm specified by the IEEE 802.15.7 standard [2]. Figure 1 shows the random access procedure for the CSMA/CA in IEEE 802.15.7 standard, which is similar to IEEE 802.15.4 [11]. In this procedure, each device initializes two variables: NB and BE; NB is the number of backoff stages which is the device tries to backoff, and BE is the backoff exponent, which is related to how many backoff periods a device shall wait before attempting to access a channel. Before a data-packet transmission, the device waits for the backoff period by counting down the backoff timer randomly selected from a range of Contention Window (CW), which is given by $2^{BE}-1$. Once the backoff timer becomes zero, the device performs CCA, in which it checks the channel state whether the other devices transmit some signals busy or not. When a device detects a channel idle at CCA, it transmits a data packet to the coordinator. When a device detects a channel busy at CCA, it increases its BE and NB. Similarly, the device increases them in the case of no acknowledgment (ACK) frame reception from the coordinator within a duration of macAckWaitDuration, where macAckWaitDuration is the maximum number of optical clocks to wait for an ACK frame to arrive following a transmitted data packet.

The hidden device problem is a well-known problem in radio CSMA/CA-based wireless networks, such as Wireless Local Area Networks (WLAN). In VLC networks, however, the hidden device collisions become a more serious problem due to the directional transmission with narrow beamwidth. In IEEE 802.15.7 VLC networks, signal collisions induced by hidden devices cause significant degradation of throughput. Figure 2 shows the mechanism of the hidden device problem in VLC networks. In the figure, Device 1 transmits a signal to the coordinator. As shown in Figure 2, the transmitted signal with high directionality only propagates within a limited direction. Therefore, Device 2 cannot detect the signal of Device 1, and then both Devices 1 and 2 send signals simultaneously, leading to signal collisions at the coordinator. These hidden device collisions due to the directionality of the transmitted signal frequently appear in the VLC networks, and thus the network performance degrades significantly. In addition, it is stated that the channel access mechanism of IEEE 802.15.7 makes the hidden device problem more serious. The channel access mechanism of IEEE 802.15.7 is designed based on IEEE 802.15.4 which is often applied to sensor networks. Generally, omni antenna is implemented to the devices for sensor networks. The channel access mechanism of IEEE 802.15.4 is also designed under the consideration of the use of omni antenna, and it is easy for each sensor device to sense signals transmitted by the neighbors. In VLC networks with CSMA/CA, on the other hand, it is difficult for each device to sense signals transmitted by the neighbors. This means the channel access mechanism of IEEE 802.15.7 and characteristics of visible light makes hidden devices problem in networks more serious.Various research has investigated the hidden device problem in IEEE 802.15.7 VLC networks [3,4,6,10,12]. Ref. [3] investigates the effect of hidden devices in uplink communication in VLC networks. Ref. [4] investigates the effect of the problem of the hidden device in IEEE 802.15.7 standard, where the authors simulate the uplink system performance using a slotted CSMA random access procedure. Ref. [12] evaluates performance in three situations: no hidden devices, a single hidden device, and no visibility between devices.In [4], the authors analyze the effect of hidden devices in VLC networks and report that the packet loss rate reaches almost 100% due to hidden device problems even under the network with low load. Therefore, the packet collisions caused by hidden devices are serious in VLC networks, and we must solve them.

### 2.2. FD-RTS/CTS Aided CSMA/CA Mechanism

One of the key solutions against the above-mentioned problem is to employ the FD communication technique [6,7,8,9,10]. FD communications essentially can mitigate the hidden device problem. Figure 3 shows the mechanism of solving hidden device problem in FD VLC networks. In this figure, Device 1 transmits a signal to the coordinator. For protecting the data signal transmitted by Device 1 from its hidden devices, the coordinator sends a busytone signal with no information data to all devices during the reception of the data signal from Device 1. By receiving the busytone signal from the coordinator, Device 2 recognizes that Device 1 is sending data, and thus it has to avoid transmitting. Therefore, hidden device collisions can be reduced through FD communications.

FD communication can be easily applied to VLC networks compared with radio communication networks. Due to the diffraction of radio signal, FD communication in radiofrequency requires self-interference cancelation, which has been one of the challenging tasks for implementation of FD in radiofrequency communication. With radio communications, the antenna is generally shared for signal transmission and reception. On the other hand, VLC devices use separate components for transmission and reception circuits, such as LED and photodetectors. In addition, the received signal strength is sufficiently small for the reflected light compared with the direct one [13]. This means that diffraction of VLC is sufficiently small that the effect of self-interference is ignored. Therefore, the interference from the signal transmitted at the same device to the receiving signal can be ignored. For this reason, it is easier to employ FD wireless communications in VLC networks than in radio communication networks.

Based on the feature of the FD communication in VLC networks, some MAC protocols have been proposed for FD VLC networks. In [6,7,10], hidden device collisions are mitigated by using a busytone signal. The authors of [6,7] evaluate the performance for IEEE 802.15.7 based FD VLC networks. Ref. [7] proposed two protocols, U-ALOHA and FD-CSMA, to avoid the hidden device problem. This work avoids hidden device collisions by treating downlink communication as a busytone signal for uplink communication.However, data packet collisions remain due to the contentions among the data-packet transmissions simultaneously from multiple devices. For suppressing wasted time due to the collisions, exchange of control frames, Request To Send/Clear To Send (RTS/CTS) is applied to FD-CSMA protocols for VLC networks in [10].

Figure 4 shows an example of channel access for the FD-RTS/CTS scheme [10] in the network with one coordinator and two devices. Due to the narrow beamwidth in the VLC communications, Device 1 cannot detect the signal transmitted from Device 2 and vice versa; that is, Devices 1 and 2 are in the hidden device relationship. The fundamental channel access mechanism for the FD-RTS/CTS scheme follows IEEE 802.15.7, as shown in Figure 1. Prior to DATA transmission, each device needs to exchange RTS/CTS frames. Before sending the RTS frame, Device 1 performs CCA when its backoff timer is zero at a time point of (a) in Figure 4. After detecting the channel idle at (a), it starts to transmit an RTS frame to the coordinator at (b). The coordinator starts to transmit a busytone signal to all the devices in the FD communication immediately after it detects the RTS frame from Device 1. Due to the busytone signal from the coordinator, Device 2 detects the channel busy at the CCAs at (b), (c), and (d). As a result, Device 1 can avoid hidden device collisions through FD communication with busytone transmission in VLC networks.

Although the FD-RTS/CTS scheme can mitigate hidden device collisions, it causes another problem. As shown in Figure 4, Device 2 keeps increasing its BE and NB values at every CCA because it detects the channel busy due to the busytone signal transmitted from the coordinator. As a result, the data packet is frequently discarded after reaching the maximum value macMaxCSMABackoffs for NB at a time point of (d). In the FD VLC networks, it is supposed that there are more opportunities of increasing BE and NB by detecting channel busy than by frame collisions. This means that the transmitting busytone to avoid hidden device collisions causes serious packet discard, thus significantly degrading the network performance. Therefore, it is necessary to propose an improved backoff scheme to suppress packet discard.

### 2.3. Performance Analysis for IEEE 802.15.7 Networks

There are several papers that clarify the behavior of the IEEE 802.15.7 network by establishing the theoretical analysis model. Such analytical models for IEEE 802.15.7 have been proposed by extending the traditional Markov-chain model proposed by Bianchi [14]. Ref. [15] proposed a priority-based MAC protocol for IEEE 802.15.7 VLC and analyzed its performance using the traditional Markov-chain model. Ref. [16] also uses the Markov-chain model to analyze the network performance under non-saturated conditions. Ref. [17] has proposed the enhanced model to consider the detail operations in IEEE 802.15.7 MAC. This model enables us to consider the duration of packet transmission and backoff duration, which is one of the important factors for considering the behavior of IEEE 802.15.7 networks. The model of [17] assumes that the probability of sensing a channel busy by a device is independent of the backoff stage and the value of the backoff counter. This assumption may cause errors between analytical results and simulation ones. In order to enhance the accuracy of the analytical model, a semi-analytic approach is used in the model of Ref. [18]. The model of Ref. [19] considers more derail operations in IEEE 802.15.7 networks by extending the model of Ref. [18]. In addition, Ref. [19] analyzes the effect of using CCA twice when using a slotted CSMA/CA random access procedure defined in the IEEE 802.15.7 standard. However, Refs. [17,18,19] assume no hidden devices and network under saturated conditions. Ref. [10] analyzed the performance of the FD-RTS/CTS scheme by extending the model of [17]. Ref. [10] assumes IEEE 802.15.7 networks with hidden devices.

## 3. Improved Backoff Scheme for FD VLC Networks

This section proposes an improved backoff scheme for FD-RTS/CTS mechanism in VLC networks. The proposed scheme aims to decrease packet discards caused by busytone detection in VLC networks with the conventional FD-RTS/CTS scheme. In the conventional backoff scheme, the devices increase the values of BE and NB when detecting the channel busy at CCA or the frame collisions after frame transmission. In FD-RTS/CTS scheme, however, the data packet is frequently discarded after reaching the maximum value macMaxCSMABackoffs for NB. In the proposed backoff scheme, therefore, the devices increase the values of BE and NB only when transmitting failure of data packets; that is, the values of BE and NB are kept as the current ones for detecting the channel busy at CCA. By using the proposed scheme, packet discards occur only for frame collisions.

As a side effect of the applied modification, the proposed scheme would cause a larger transmission delay. This is because it always keeps the BE and NB for the detecting of channel busy at CCA. Therefore, to suppress the larger transmission delay, a limitation for keeping the BE and NB is applied to the proposed scheme. Applying a limitation for keeping the BE and NB would provide a small access delay even under the low packet discard probability.

Figure 5 shows the random access procedure for the proposed backoff scheme. Here, let *k* and *K* be the number of times to keep the BE and NB in the same backoff stage and its limitation (maximum value), respectively. The device keeps the BE and NB for k≤K. For *k* over *K*, the device increases the number of backoff stages. That is, for K=0, the device does not keep BE and NB, corresponding to the same behavior as the FD-RTS/CTS scheme.

Figure 6 shows an example of channel access for the proposed backoff scheme. From Figure 4 and Figure 6, the FD-RTS/CTS scheme increases the values of BE and NB for every detection of channel busy at CCA, whereas the proposed scheme keeps these values. Although the modification in the proposed scheme from the original backoff procedure in IEEE 802.15.7 is quite simple, the proposed scheme significantly decreases packet discards in FD VLC networks.

## 4. Theoretical Analysis of the Improved Backoff Scheme

This section provides the performance analysis of the proposed scheme through a mathematical approach.

### 4.1. Assumption for the Analysis

The analysis in this paper is based on the following assumptions:The network is star topology with one coordinator and *N* devices. Each device cannot detect any signal transmitted from other ones due to a hidden device relationship.Each device generates data packets whose size is *P* [bytes], and the destination is the coordinator. All the devices have the same packet arrival rate λ [1/s] following a Poisson arrival process.Network offered load is *O* [Mbps]. Therefore, the offered load of each device is $\frac{O}{N}$ [Mbps]. This means $\lambda =\frac{O}{NP}$.Because the coordinator transmits a busytone signal during the reception of a signal from the devices, no transmission failure induced by the hidden device occurs. When multiple devices start to transmit RTS frames simultaneously, they fail their transmission due to signal collision.

### 4.2. Markov-Chain Model for the Proposed Backoff Scheme and the Overview of Calculation

There are many studies to represent the behavior of a CSMA/CA-based network, such as IEEE 802.11 WLAN and sensor networks with IEEE 802.15.4, based on the Markov-chain model [14,20,21,22,23]. The most famous model is proposed by [14], and theoretical analysis for IEEE 802.15.7 has been proposed in [10,15,16,17,18,19] by extending the Markov-chain model of [14]. The Markov-chain model is one of the most powerful mathematical tools providing a simple numerical performance analysis scheme. The FD-RTS/CTS scheme [10] also uses the Markov-chain model to analyze its MAC protocol. This paper provides a Markov-chain model for channel access with the proposed backoff scheme.

Figure 7 shows the Markov-chain model for the proposed backoff scheme. This analytical model also follows the basic idea of the Markov-chain model analysis, which is introduced for IEEE 802.11 networks in [14]. In order to consider the operation of the proposed backoff procedure as shown in Figure 5, however, a three-dimensional Markov-chain model is applied. The state i,j,k in the Markov-chain model represents a state for a device with the backoff-timer of *j* after keeping BE and NB *k* times on the *i*-th backoff stage. In Figure 7, the range of contention window on the *i*-th backoff stage is
(1)$$W_{i}=2^{\min\left(macMinBE+i,macMaxBE\right)},$$
where macMinBE and macMaxBE are the minimum, and maximum values of BE, respectively. *m* is the value of macMaxCSMABackoffs. In the Markov-chain model, $p_{c}$ and α are “collision probability” and “channel busy probability”, respectively. The definition $p_{c}$ is the probability that an RTS frame transmitted from a device collides with the other RTS frames. The definition of α is the probability that a device detects channel busy at CCA. The expression for these key parameters are derived in Section 4.6. To consider the backoff operation in IEEE 802.15.7 FD VLC networks, the Markov-chain model consists of three periods; backoff period, RTS/CTS period, and transmission period. Specifically, the states for i∈(0,m), $j\in (0,W_{i}-1)$, and k∈(0,K), those for i∈(0,m) and $j\in \{RTS,SIFS_{1},CTS,SIFA_{2}\}$, and those for (−1,j) for j∈(1,B) belong to the backoff period, the RTS/CTS period, and the transmission period, respectively. Here, *B* is the successful DATA packet transmission duration of
(2)$$\begin{array}{c} B=T_{\mathrm{DATA}}+T_{\mathrm{SIFS}}+T_{\mathrm{ACK}}+T_{\mathrm{LIFS}}, \end{array}$$
where $T_{\mathrm{SIFS}}$ and $T_{\mathrm{LIFS}}$ are durations of Short Inter Frame Space (SIFS) and Long Inter Frame Space (LIFS), and $T_{\mathrm{DATA}}$ and $T_{\mathrm{ACK}}$ are durations for transmitting a DATA packet and an ACK frame, respectively.

If the channel is idle at CCA after the backoff timer reaches zero, the device enters the RTS/CTS period. The states (i,RTS) and (i,CTS) for i∈(0,m) mean that a device transmits an RTS frame and a CTS frame, respectively. The states $\left(i,SIFS_{1}\right)$ and $\left(i,SIFS_{2}\right)$ for i∈(0,m) mean that the device waits for SIFS period after the RTS-frame transmission and CTS-frame transmission, respectively. If the device successfully receives the CTS frame transmitted from the coordinator, it enters the transmission period. If not, the device enters the backoff period with increased *i*.

With the Markov-chain model, we first derive a CCA probability φ for a given packet arrival rate by considering the packet-existence probability *q* derived from queuing theory, which is given in Section 4.5. The theoretical expressions for collision probability $p_{c}$ and channel busy probability α are derived by using the CCA probability φ, which is given in Section 4.6. Finally, we can numerically solve these three probabilities, φ, $p_{c}$, and α, using the Newton method. By substituting the obtained solutions into the expressions for throughput *S*, packet-discard probability $p_{d}$, and access delay $D_{mac}$, we can finally obtain theoretical values for *S*, $p_{d}$, and $D_{mac}$ associated with the packet arrival rate, which is given in Section 4.7 and Section 4.8.

### 4.3. Transition Probability for the Proposed Backoff Scheme

The transition probabilities related to the above RTS operations in the RTS/CTS period in the Markov-chain model are expressed as
(3)$$\begin{array}{ccccc} \; & P\{i,SIFS_{1}|i,RTS\}=1, & i\in (0,m),\\ \; & P\{i,CTS|i,SIFS_{1}\}=1, & i\in (0,m),\\ \; & P\{i,SIFS_{2}|i,CTS\}=1, & i\in (0,m),\\ \; & P\left\{i,j,0|i-1,SIFS_{2}\right\}=\frac{p_{c}}{W_{i}}, & i\in (1,m), & j\in (0,W_{i}-1), & \\ \; & P\left\{-1,1|i,SIFS_{2}\right\}=1-p_{c}, & i\in (0,m), \end{array}$$
where P{x|y} is the transition probability from state *y* to state *x*. The transition probability in the transmission period is
(4)$$\begin{array}{ccc} P\{-1,j+1|-1,j\}=1, & \; & j\in (1,B-1). \end{array}$$

Following the original backoff procedure in IEEE 802.15.7, which is explained in Section 2.1, the transition probabilities regarding the backoff period in the proposed backoff scheme are expressed as
(5)$$\begin{array}{cccccc} \; & P\{i,j,k|i,j+1,k\}=1, & i\in (0,m), & j\in (0,W_{i}-2), & k\in (0,K),\\ \; & P\{i,RTS|i,0,k\}=1-\alpha , & i\in (0,m), & \; & k\in (0,K),\\ \; & P\left\{0,j,0|m,SIFS_{2}\right\}=\frac{p_{c}}{W_{0}}, & \; & j\in (0,W_{0}-1), & \; & \\ \; & P\left\{0,j,0|m,0,K\right\}=\frac{\alpha }{W_{0}}, & \; & j\in (0,W_{0}-1), & \; & \\ \; & P\left\{0,j,0|-1,B\right\}=\frac{1}{W_{0}}, & \; & j\in (0,W_{0}-1). \end{array}$$

The transition regarding the procedure to keep BE and NB in the proposed scheme is expressed as
(6)$$\begin{array}{ccccc} \; & P\left\{i,j,k|i,0,k-1\right\}=\frac{\alpha }{W_{i}}, & i\in (0,m), & j\in (0,W_{i}-1), & k\in (1,K),\\ \; & P\left\{i,j,0|i-1,0,K\right\}=\frac{\alpha }{W_{i}}, & i\in (1,m), & j\in (0,W_{i}-1). \end{array}$$

### 4.4. Stationary Distribution Probability in the Markov-Chain Model

We define the stationary distribution of the states in the backoff period of the Markov-chain model as $b_{i,j}$, for i∈(0,m), $j\in (0,W_{i}-1)$. From the transition probabilities in Equations (Equation 3)–(Equation 5), we obtain the following relational equation:(7)$$b_{i,j,k}=\frac{(W_{i}-j)\alpha }{W_{i}}\cdot b_{i,0,k-1}.$$

By substituting j=0 in Equation (Equation 7), $b_{i,0}$ is rewritten as
(8)$$\begin{array}{c} b_{i,0,k}=\alpha \cdot b_{i,0,k-1}=\cdots =\alpha^{k}\cdot b_{i,0,0}. \end{array}$$

From Equations (Equation 7) and (Equation 8), we obtain
(9)$$b_{i,j,k}=\frac{\left(W_{i}-j\right)}{W_{i}}\cdot b_{i,0,k}.$$

If the device keeps the BE and NB continuously K+1 times at the same backoff stage or a collision occur, the *i* is increased by 1, namely
(10)$$b_{1,0,0}=\alpha b_{0,0,K}+p_{c}b_{0,SIFS_{2}}=Xb_{0,0,0},$$
where
(11)$$X=\alpha^{K+1}+p_{c}\left(1-\alpha^{K+1}\right).$$

From Equation (Equation 10), we obtain
(12)$$b_{i,0,0}=Xb_{i-1,0,0}=\cdots =X^{i}b_{0,0,0},\;i\in \left(0,m\right).$$

Stationary distributions for the RTS/CTS period and the transmission period satisfy the following relation equations: (13)$$\begin{array}{cc} b_{i,j}=\left(1-\alpha \right)\sum_{k=0}^{K}b_{i,0,k}=X^{i}\left(1-\alpha^{K+1}\right)b_{0,0,0}, & \mathrm{for}\;i\in \left(0,m\right),j\in \left\{RTS,SIFS_{1},CTS,SIFA_{2}\right\}, \end{array}$$
and
(14)$$\begin{array}{cc} b_{-1,j}=\sum_{i=0}^{m}\left(1-p_{c}\right)b_{i,SIFS_{2}}=\left(1-\alpha^{K+1}\right)\left(1-p_{c}\right)\frac{1-X^{m+1}}{1-X}b_{0,0,0}, & \mathrm{for}\;j\in (1,B), \end{array}$$
respectively. Because the sum of the stationary distribution probability of all the state should be one, we have
(15)$$1=\sum_{i=0}^{m}\sum_{j=0}^{W_{i}-1}\sum_{k=0}^{K}b_{i,j,k}+A\sum_{i=0}^{m}b_{i,-1}+B\cdot b_{-1,1}.$$

By substituting Equations (Equation 9) and (Equation 12)–(Equation 14) for Equation (Equation 15), $b_{0,0}$ can be obtained as
(16)$$\frac{1}{b_{0,0,0}}=\frac{1-\alpha^{K+1}}{1-\alpha }\sum_{i=0}^{m}\frac{W_{i}+1}{2}X^{i}+A\left(1-\alpha^{K+1}\right)\frac{1-X^{m+1}}{1-X}+B\left(1-p_{c}\right)\left(1-\alpha^{K+1}\right)\frac{1-X^{m+1}}{1-X}.$$
where *A* is the time taken in the RTS/CTS period of
(17)$$\begin{array}{c} A=T_{\mathrm{RTS}}+T_{\mathrm{SIFS}}+T_{\mathrm{CTS}}+T_{\mathrm{SIFS}}, \end{array}$$
where $T_{\mathrm{RTS}}$ and $T_{\mathrm{CTS}}$ are the durations for transmitting RTS and CTS frames, respectively. From Equation (Equation 16), each stationary distribution probability can be expressed as a function of $p_{c}$ and α.

### 4.5. CCA Probability φ

This section derives the expression for CCA probability φ, which is the probability that a device performs CCA. When a device has at least one data packet in its buffer, it attempts to transmit a data packet. Here, we define the packet-existence probability *q* as the probability that a device has at least one packet to transmit. In addition, we define $\varphi^{\prime }$ as the probability that the device, whose buffer has always at least one packet, performs CCA when its backoff timer reaches zero. Then, φ is expressed as the product of *q* and $\varphi^{\prime }$:(18)$$\begin{array}{c} \varphi =q\varphi^{\prime }. \end{array}$$

Following the explanations, we derive expressions of $\varphi^{\prime }$ and *q*, respectively. When a device always has at least one packet in its buffer, q=1 is satisfied. Namely, this condition is called a saturated condition. By calculating the sum of the stationary distributions for state {i,0,k} for i∈(0,m) and k∈(0,K) in the Markov-chain model, therefore, the probability that a backoff timer of the device is zero under the saturated condition is obtained as
(19)$$\begin{array}{c} \varphi^{\prime }=\frac{\sum_{i=0}^{m}\sum_{k=0}^{K}b_{i,0,k}}{\sum_{i=0}^{m}\sum_{j=0}^{W_{i}-1}\sum_{k=0}^{K}b_{i,j,k}}=\frac{\frac{1-\alpha^{K+1}}{1-\alpha }\frac{1-X^{m+1}}{1-X}b_{0,0,0}}{\frac{1-\alpha^{K+1}}{1-\alpha }\sum_{i=0}^{m}\frac{W_{i}+1}{2}X^{i}b_{0,0,0}}=\frac{1-X^{m+1}}{\left(1-X\right)\sum_{i=0}^{m}\frac{W_{i}+1}{2}X^{i}}. \end{array}$$

From the queuing theory, *q* is obtained as a ratio of the frame arrival rate λ to the frame service rate μ, namely
(20)$$\begin{array}{c} q=\min\left(\frac{\lambda }{\mu },1\right). \end{array}$$

In Equation (Equation 20), the function of min prevents *q* exceeding 1. Since the Markov-chain model includes the three periods for one data packet transmission, the inverse of μ corresponds to the access delay $D_{mac}$ for a data packet. From the characteristics of the Markov-chain model, the access delay is derived as the expected recurrence time of the state (−1,B), which is given by the inverse of the stationary probability:(21)$$\begin{array}{c} D_{mac}=\frac{1}{b_{-1,B}}=\frac{1}{\left(1-p_{c}\right)\left(1-\alpha^{K+1}\right)\frac{1-X^{m+1}}{1-X}b_{0,0,0}}=\frac{1}{\mu }. \end{array}$$

Therefore, by substituting Equations (Equation 19)–(Equation 21) in Equation (Equation 18), φ can be derived as
(22)$$\begin{array}{c} \varphi =\min\left(\frac{\lambda }{\left(1-p_{c}\right)\left(1-\alpha^{K+1}\right)\frac{1-X^{m+1}}{1-X}b_{0,0,0}},1\right)\cdot \frac{1-X^{m+1}}{\left(1-X\right)\sum_{i=0}^{m}\frac{W_{i}+1}{2}X^{i}}. \end{array}$$

### 4.6. Collision Probability $p_{c}$ and Channel Busy Probability α

From the Assumption 4, frame collisions occur when multiple devices start to transmit simultaneously. Therefore, the collision probability $p_{c}$ is expressed as
(23)$$\begin{array}{c} p_{c}=1-{\left(1-\varphi \right)}^{N-1}. \end{array}$$

Because the coordinator transmits a busytone signal while at least one device transmits an RTS frame or a data packet, a device detects a busy channel at CCA if the other devices transmit an RTS frame or a data packet. The probability that at least one device transmits an RTS frame or a data packet is expressed as $1-{\left[1-\varphi (1-\alpha )\right]}^{N-1}$. The expected duration for one data packet transmission with RTS/CTS exchange is given by $A+B(1-p_{c})$. Because the channel busy probability depends on the duration of data-packet transmission [10], α is expressed as
(24)$$\begin{array}{c} \alpha =\left[A+B(1-p_{c})\right]\left\{1-{\left[1-\varphi (1-\alpha )\right]}^{N-1}\right\}. \end{array}$$

By solving Equations (Equation 22)–(Equation 24) for arbitrary value of λ numerically, we can get the solution for φ, $p_{c}$, and α.

### 4.7. Packet-Discard Probability $p_{d}$

Packet-discard probability $p_{d}$ is the probability that a device discards a packet. In the proposed scheme, a device discards a data packet when the backoff stage is increased m+1 times. Therefore, the packet-discard probability $p_{d}$ is expressed as
(25)$$\begin{array}{ccc} p_{d} & = & X^{m+1}\\ & = & {\left[\alpha^{K+1}+p_{c}\left(1-\alpha^{K+1}\right)\right]}^{m+1}. \end{array}$$

### 4.8. Throughput *S*

Throughput *S* is defined as the sum of throughput of each device in the network. Here, we define *s* as the throughput for one device. When the network is in a non-saturated condition, which is q<1; it is stated that almost all of the data packets generated at a device are successfully sent to the coordinator. Considering the discarded data packet, therefore, the throughput for a device under non-saturated conditions is expressed as
(26)$$\begin{array}{cc} s=\lambda (1-p_{d})P, & for\;q<1. \end{array}$$

On the other hand, each device always has data packets in its buffer in the saturated condition. Therefore, the throughput under the saturated condition is expressed as the ratio of one data payload size to the access delay. Considering the discarded data packet, therefore, the throughput of a device under saturated condition is expressed as
(27)$$\begin{array}{cc} s=\frac{(1-p_{d})P}{D_{mac}}, & for\;q=1. \end{array}$$

By considering Equations (Equation 20), (Equation 26), and (Equation 27), *s* is rewritten as
(28)$$\begin{array}{c} s=\frac{q(1-p_{d})P}{D_{mac}}. \end{array}$$

Therefore, the network throughput *S* is obtained as S=N×s.

## 5. Performance Evaluation and Analysis of the Improved Backoff Scheme

This section demonstrates the effectiveness of the proposed backoff scheme through computer simulation and theoretical analysis. Table 1 shows major system parameters and their settings. We developed the original network simulator constructed in C language. In this simulator, MAC-layer operations, which are described in Figure 5, are implemented. For evaluating MAC layer performance, in the simulation, it is assumed the ideal channel condition in the PHY layer is similar to the theoretical analysis in Section 4. The other assumptions related to network topology and traffic are also the same as the theoretical analysis.

In Figure 8, Figure 9, Figure 10 and Figure 11, we also show the analytical and simulation results for the conventional FD-RTS/CTS scheme for performance comparison. The analytical results for the FD-RTS/CTS can be given by the similar calculation as that explained in Section 4 with replacing Equation (Equation 6) to $P\left\{i+1,j|i,0\right\}=\frac{\alpha }{W_{i+1}}$. For discussing appropriate *K* by using analytical model, on the other hand, Figure 12, Figure 13 and Figure 14 include only the analytical results.

### 5.1. Performance Evaluation of the Improved Backoff Scheme

Figure 8, Figure 9 and Figure 10 show the packet discard probability, network throughput, and access delay as a function of offered load. In these figures, the number of devices *N* is 5, and the limitation for keeping BE and NB *K* is an infinite value. From Figure 8, the proposed scheme achieves almost no packet discard for any offered load. On the other hand, the packet discards frequently occur for any offered load in the FD-RTS/CTS scheme. It is confirmed from Figure 8 that the packet discard probability significantly improves for all offered loads by applying the proposed scheme. As shown in Figure 9, the proposed scheme improves network throughput when 10<O<40 [Mbps]. This is because the proposed scheme decreases the packet discard under non-saturated conditions. Therefore, the effectiveness of the proposed scheme can be confirmed. However, it is seen from Figure 10 that the access delay for the proposed scheme is longer than that for the FD-RTS/CTS scheme. This result shows that the proposed scheme reduces packet discard, although it degrades the access delay.

To confirm the effectiveness of operation keeping BE and NB in the proposed scheme, we evaluate the proposed and FD-RTS/CTS schemes under the same condition of no packet discard. Figure 11 shows the access delay for N=5, K=∞, and the maximum backoff stage m=1000. Setting *m* as an extremely large value can evaluate access delay for FD-RTS/CTS schemes with approximately zero packet discard probability. It is seen from Figure 11 that the access delay for the proposed scheme is smaller than that for the FD-RTS/CTS scheme. This is because the proposed scheme tends to choose a smaller backoff timer than the FD-RTS/CTS scheme. Therefore, it can be seen that the proposed scheme is more effective than the FD-RTS/CTS scheme.

### 5.2. Appropriate Value of *K*

The proposed scheme reduces the packet discards, whereas it increases the access delay. This is because the proposed scheme keeps BE and NB. Therefore, we evaluate the performance for varying the *K* in order to find an appropriate value for *K*. Figure 12, Figure 13 and Figure 14 show the packet discard probability, access delay, and network throughput for K=0,1,5, and 30 as a function of network offered load. From Figure 12 and Figure 13, it can be seen that the packet discard probability is smaller and the access delay is larger with increasing *K*. On the other hand, from Figure 14, we can see that, when setting K=5, the network throughput is almost the same as that for K=30. Therefore, the proposed scheme can achieve high throughput without constantly keeping BE and NB.

In order to achieve higher network throughput, it is necessary to reduce the packet discard. However, reducing packet discards causes increased access delay, which may be unacceptable to some applications. Therefore, it is useful to achieve an appropriate *K* which satisfies an access delay specified by a given application. Table 2 shows the value of *K* and the gain for the maximum throughput under an access delay limitation of 50 μs. Here, the network offered load is set to 15 Mbps, which is under a non-saturated condition. As shown in Table 2, the network throughput of the FD-RTS/CTS scheme (corresponding to the proposed scheme with K=0) is quite lower than the network offered load. This is because the device frequently discards packets, even in non-saturated conditions. On the other hand, the network throughput of the improved backoff scheme is higher than that of the FD-RTS/CTS scheme, satisfying the specified access delay, which is 50 μs. As shown in Table 2, the packet discard probability increases as increasing *N*. However, more than 30 % of packets are discarded even in the non-saturated condition (15 Mbps) in the FD-RTS/CTS scheme (K=0). It is confirmed that the proposed scheme can achieve the required access delay, low packet discard probability, and high network throughput even when the number of devices is large.By selecting an appropriate value of *K*, the proposed scheme can achieve a higher network throughput while satisfying the required access delay. Acceptable packet discard probability and access delay depend on an application on networks. Through the calculation results from the mathematical model, we can obtain appropriate *K*, which satisfies the performance required for a corresponding application.

## 6. Conclusions

This paper proposed an improved backoff scheme to reduce the serious packet discard in VLC networks. In the proposed backoff scheme, each device increases the BE and NB values only for transmission failures of data packets. However, the proposed scheme reduces the packet discard while it increases the access delay. For the increased access delay, we proposed to apply the limitation for keeping the BE and NB. We also presented its performance based on the theoretical analysis. The simulation and theoretical analysis results showed that the proposed backoff scheme achieved both a small packet discard probability and a small access delay under non-saturated conditions by selecting an appropriate value *K*.

## Figures and Tables

**Figure 1 sensors-21-08263-f001:**
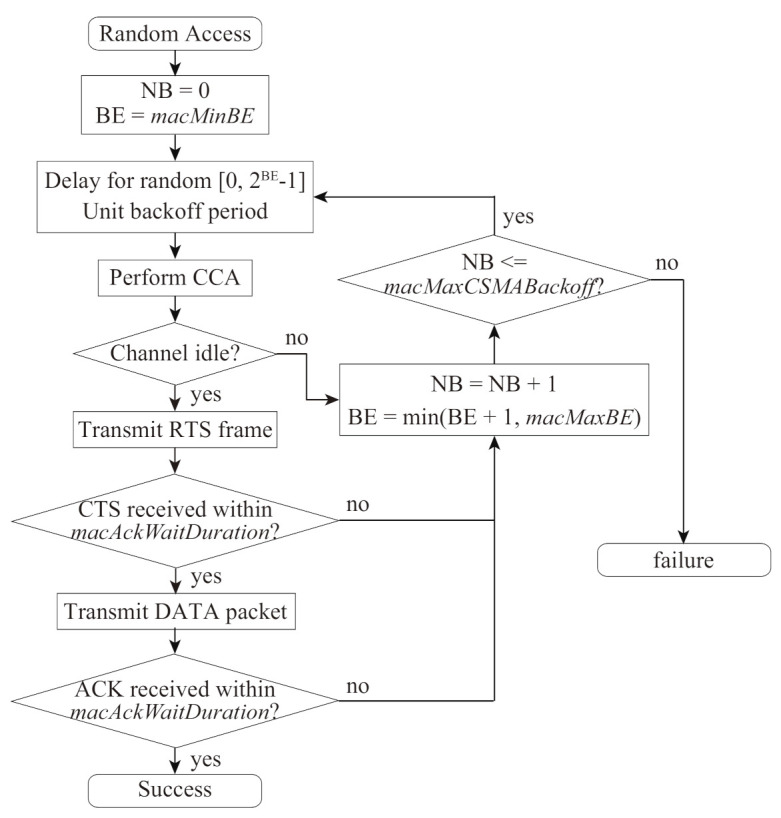
Random–access procedure for the CSMA/CA in IEEE 802.15.7 standard.

**Figure 2 sensors-21-08263-f002:**
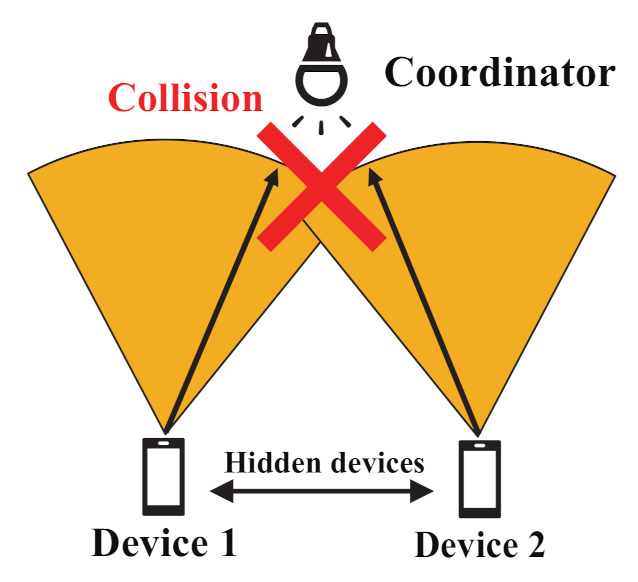
The mechanism of the hidden device problem in VLC networks.

**Figure 3 sensors-21-08263-f003:**
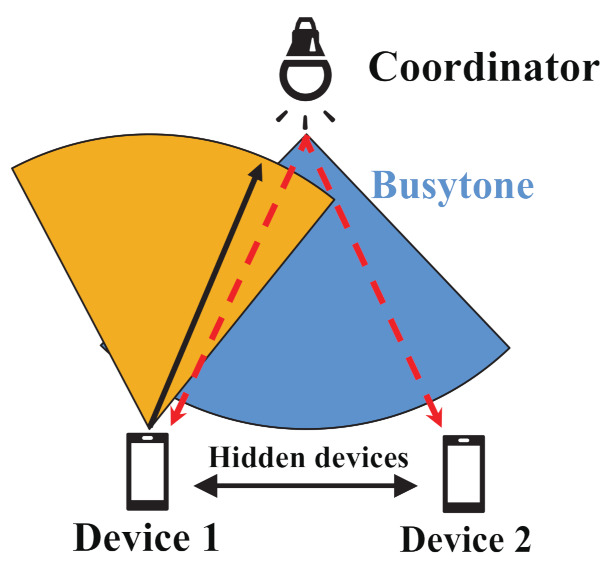
The mechanism of solving the hidden device problem in FD VLC networks.

**Figure 4 sensors-21-08263-f004:**
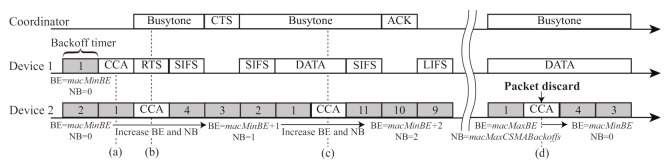
Channel access example of the FD-RTS/CTS scheme.

**Figure 5 sensors-21-08263-f005:**
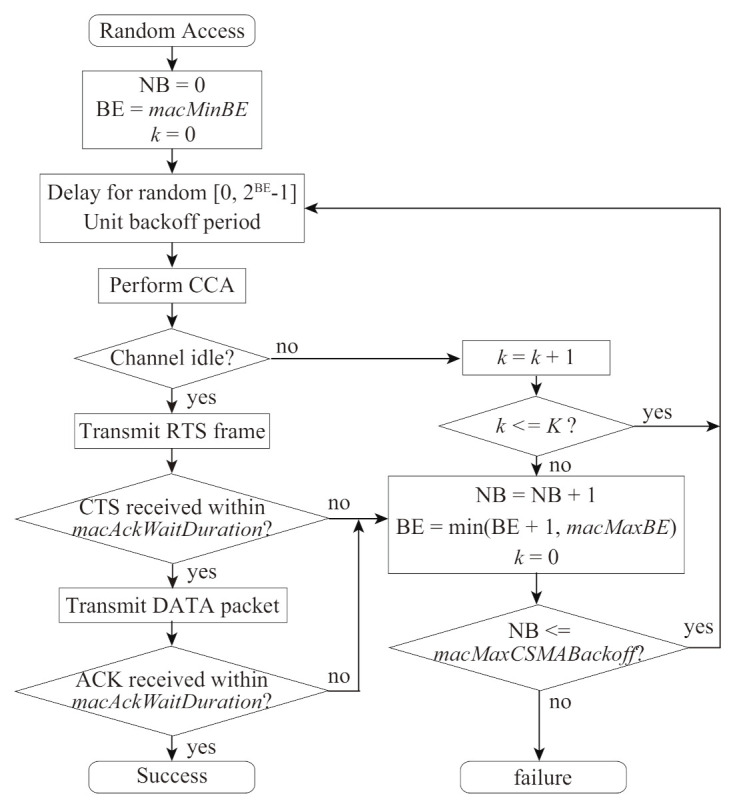
Random–access procedure for the proposed backoff scheme.

**Figure 6 sensors-21-08263-f006:**
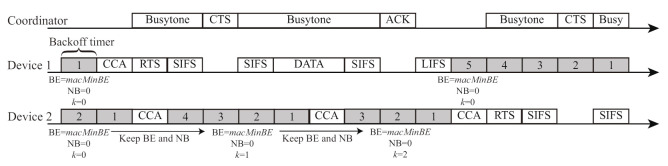
Channel access example of the proposed scheme.

**Figure 7 sensors-21-08263-f007:**
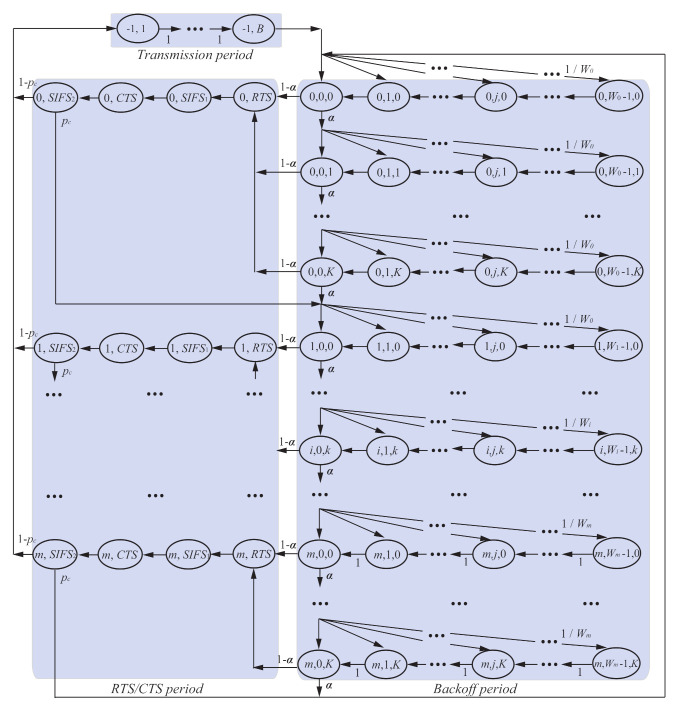
Markov–chain model for the improved backoff scheme.

**Figure 8 sensors-21-08263-f008:**
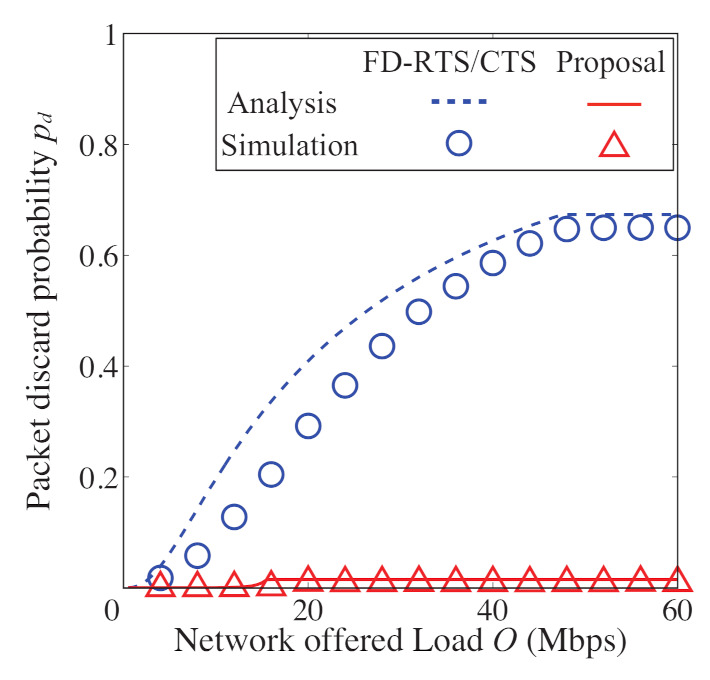
Packet discard probability as a function of offered load for N=5.

**Figure 9 sensors-21-08263-f009:**
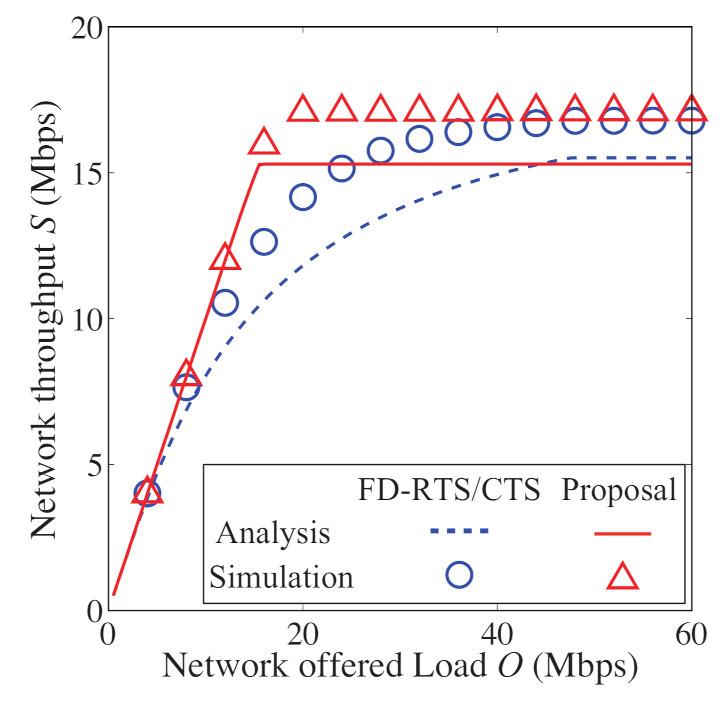
Network throughput as a function of offered load for N=5.

**Figure 10 sensors-21-08263-f010:**
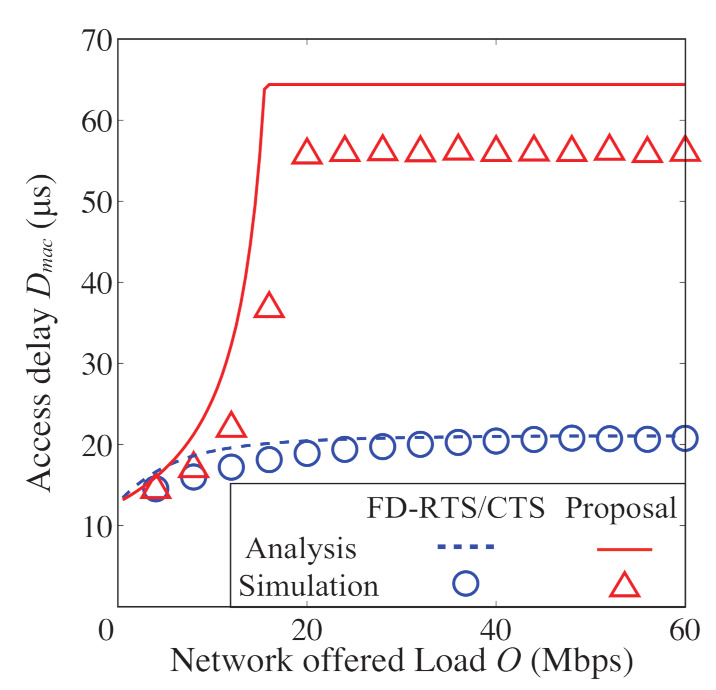
Access delay as a function of offered load for N=5.

**Figure 11 sensors-21-08263-f011:**
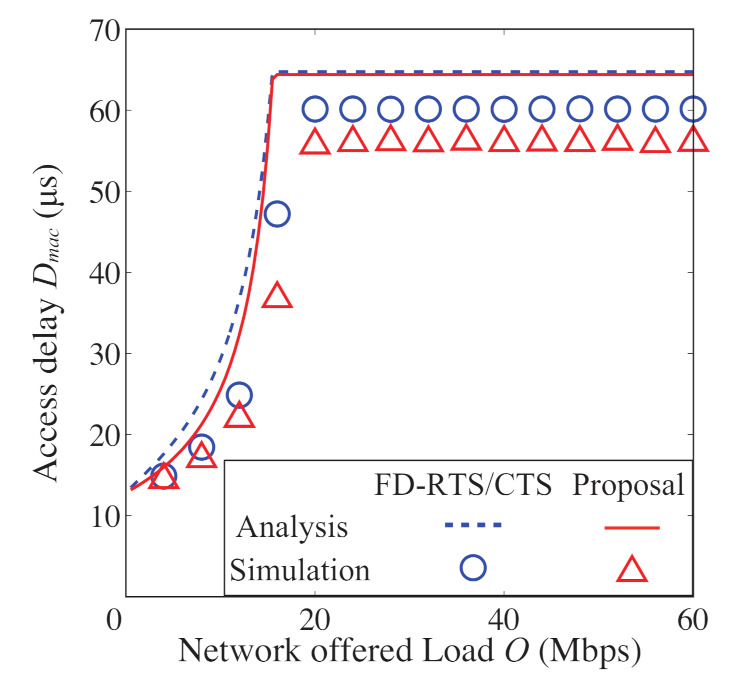
Access delay as a function of offered load, the case of no packet discard.

**Figure 12 sensors-21-08263-f012:**
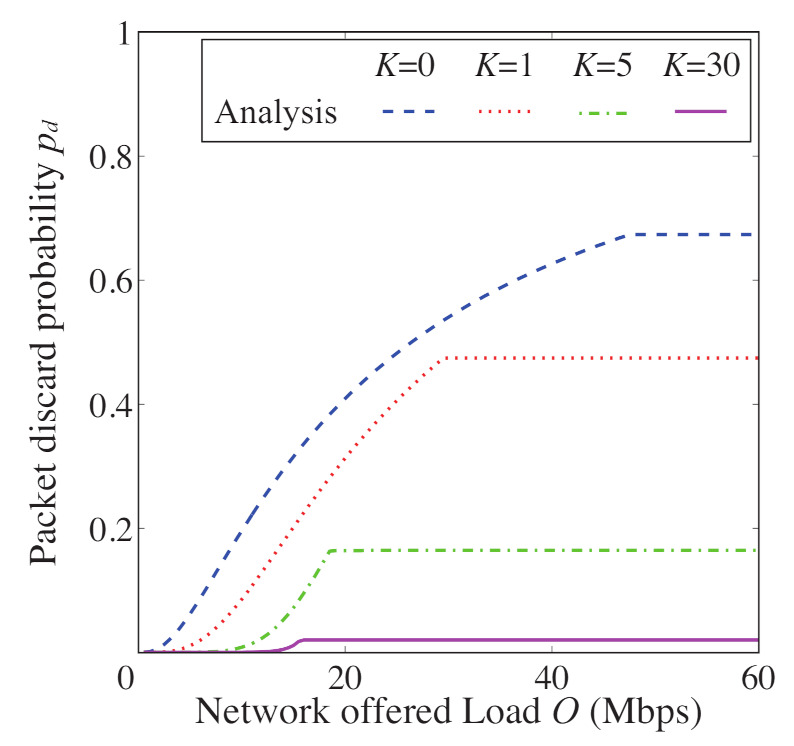
Packet discard probability as a function of offered load for N=5 and K=0,1,5, and 30.

**Figure 13 sensors-21-08263-f013:**
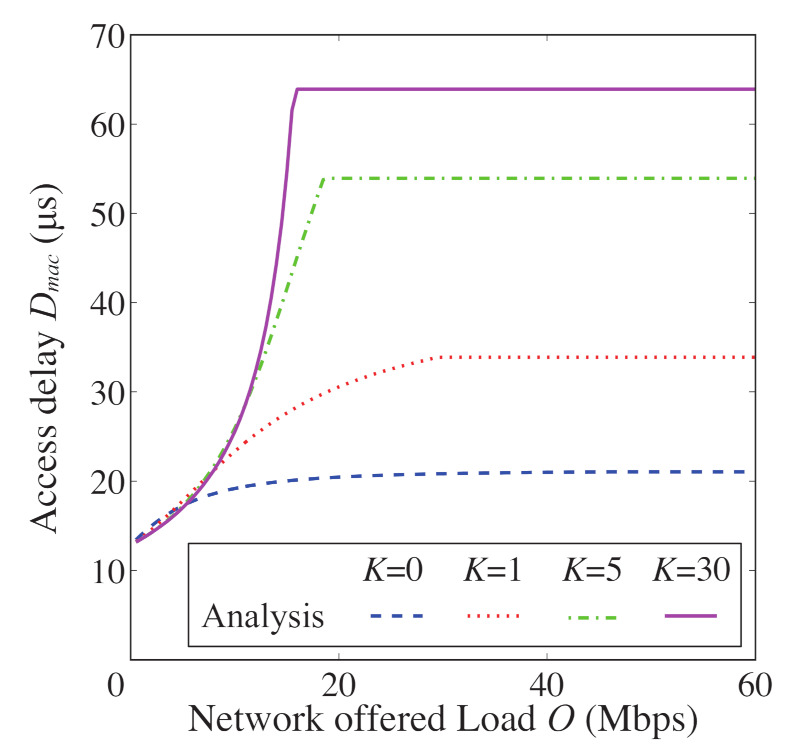
Access delay as a function of offered load for N=5 and K=0,1,5, and 30.

**Figure 14 sensors-21-08263-f014:**
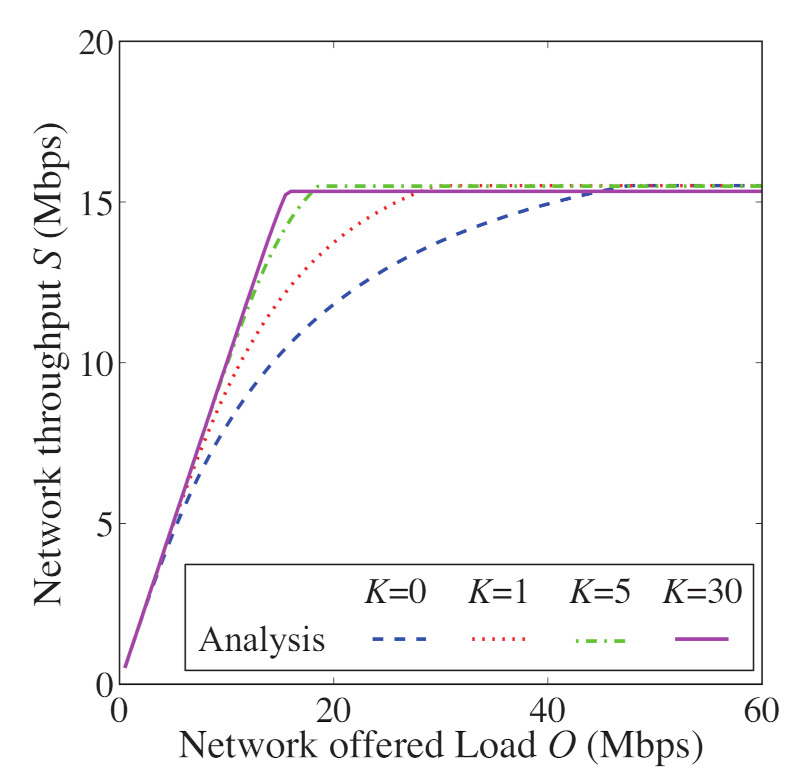
Network throughput as a function of offered load for N=5 and K=0,1,5, and 30.

**Table 1 sensors-21-08263-t001:** System parameters.

Parameter	Value
Data rate	24 Mbps
Data payload size (*P*)	50 bytes
LIFS time ($T_{\mathrm{LIFS}}$)	40 optical clocks
SIFS time ($T_{\mathrm{SIFS}}$)	20 optical clocks
RTS time ($T_{\mathrm{RTS}}$)	20 optical clocks
CTS time ($T_{\mathrm{CTS}}$)	20 optical clocks
ACK time ($T_{\mathrm{ACK}}$)	20 optical clocks
CCA time	20 optical clocks
Slot time	20 optical clocks
Optical clock rate	60 MHz
macMinBE	3
macMaxBE	5
macMaxCSMABackoffs	4

**Table 2 sensors-21-08263-t002:** Network throughput as a function of offered load for N=5 and K=0,1,5, and 30.

		Network	Packet Discard	Network	Packet Discard	
*N*	max *K*	throughput	Probability	throughput	Probability	Gain
		for max *K*	for max *K*	for *K* = 0	for *K* = 0	
5	12	14.58 Mbps	2.769%	10.27 Mbps	31.55%	1.42
10	5	13.27 Mbps	11.56%	9.71 Mbps	35.28%	1.37
15	4	12.78 Mbps	14.80%	9.53 Mbps	36.44%	1.34

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
