# Peer review of "An Improved Backoff Scheme and Its Performance Analysis for Full Duplex MAC Protocols in VLC Networksâ€"

_sensors, 2021, doi:10.3390/s21248263_

Round 1
Reviewer 1 Report
The paper with title “An Improved Backoff Scheme and its Performance Analysis for Full Duplex MAC protocols in VLC Networks” proposes an improved backoff scheme and its performance analysis to suppress the packet discard, typical when employ a full-duplex transmission technique in order to mitigate the hidden device collisions. This backoff scheme increases the Backoff Exponent (BE) and the Number of Backoff stage (NB) in IEEE 802.15.7 only when the data packet transmission fails. This scheme reduces the packet discards, whereas it increases the access delay. This increase may be unacceptable to some applications, so the authors added an access delay limitation of 50 [µs] in order to calculate the value of K and the gain for the maximum throughput.
The paper is interesting and is, in general, professionally written, but there are some parts that could be improved or rewritten.
In section 1. Introduction, in the first paragraph, authors say that “However, signal collision 21 induced by hidden devices causes more serious performance degradation than the network using radiofrequency. This is because both transmitters and receivers have narrow directional characteristics in VLC networks [3,4].” But, in the references do not appear any statement which links VLC networks with more serious performance degradation than the RF network. Could you explain in depth the relationship between VLC networks and increased degradation by hidden nodes?
In section 2. Related work, subsection 2.1. IEEE 802.15.7 CSMA/CA and Hidden device problem in VLC networks, in the first paragraph, the authors state that “In VLC networks, however, the hidden device collisions become a more serious problem due to the directional transmission with narrow beamwidth.” It is the same issue than the previous question, could you affirm that this sentence is always true? because there are many RF networks and some of them work with directive or very directive beamwidths.
In section 2. Related work, subsection 2.2. FD-RTS/CTS aided CSMA/CA Mechanism, in the second paragraph the authors say that “For this reason, it is easier to employ FD wireless communications in VLC networks than in radio communication networks.” Could you explain with detail why is better FD wireless communications in VLC networks than in radio communication networks?
In section 6. Conclusion the authors explain the improvements of this paper but, in my opinion, the authors could report that the system works correctly for a reduced number of devices connected simultaneously (N = 5) if, for example, this number increases (N = 10), the percentage of discards is multiplied by 5 even reducing the value of K.
Author Response
(1-1)
In section 1. Introduction, in the first paragraph, authors say that“ However, signal collision induced by hidden devices causes more serious performance degradation than the network using radiofrequency. This is because both transmitters and receivers have narrow directional characteristics in VLC networks [3,4]. ”But, in the references do not appear any statement which links VLC networks with more serious performance degradation than the RF network. Could you explain in depth the relationship between VLC networks and increased degradation by hidden nodes?
Answer for (1-1)
In radiofrequency networks with omni antenna, the transmitted signal propagates in all directions around the antenna. Therefore, the frame collisions induced by the hidden-device problem occur when the devices are deployed at outside of the communication range of the transmitted signal. On the other hand, in VLC networks, the signal propagates in a limited direction due to the directional transmission with narrow beamwidth. Therefore, it is difficult for VLC-network devices to detect signals transmitted by the neighbors even if the distance between the devices is short. For this reason, the effect induced by hidden devices becomes more serious problem. A similar explanation is stated in [4].
As the reviewer’s point out, there is no statement which links VLC networks with more serious performance degradation “than radiofrequency network”. Therefore, we removed “than the network using radiofrequency” from the statement in the revised manuscript. We added the explanations regarding the effects of hidden device collisions for radiofrequency and VLC networks, respectively, in Section 1. This is related to comment 1-2.
(1-2)
In section 2. Related work, subsection 2.1. IEEE 802.15.7 CSMA/CA and Hidden device problem in VLC networks, in the first paragraph, the authors state that “ In VLC networks, however, the hidden device collisions become a more serious problem due to the directional transmission with narrow beamwidth.”It is the same issue than the previous question, could you affirm that this sentence is always true? because there are many RF networks and some of them work with directive or very directive beamwidths.
Answer for (1-2)
The channel access mechanism of IEEE 802.15.7 is designed based on IEEE 802.15.4 which is often applied to sensor networks. Generally, an omni antenna is implemented to devices for sensor networks. The channel access mechanism of IEEE 802.15.4 is also designed to consider the use of omni antenna, and it is easy for each sensor device to sense signals transmitted by the neighbors. (It seems that few cases use a directional antenna in sensor networks from the viewpoint of implementation cost.) In VLC networks with CSMA/CA, on the other hand, it is difficult for each device to sense the signal transmitted by the neighbors. This means the channel access mechanism of IEEE 802.15.7 and characteristics of visible light makes hidden devices problem in networks more serious. From the above explanation, We affirm that the hidden device collisions become a more serious problem due to the directional transmission with narrow beamwidth. The related work of Ref. [4] also points out similar problems.
We added the above explanation in Section 2.1.
(1-3)
In section 2. Related work, subsection 2.2. FD-RTS/CTS aided CSMA/CA Mechanism, in the second paragraph the authors say that “ For this reason, it is easier to employ FD wireless communications in VLC networks than in radio communication networks. ” Could you explain with detail why is better FD wireless communications in VLC networks than in radio communication networks?
Answer for (1-3)
Due to the diffraction of radio signal, FD communication in radiofrequency requires self-interference cancelation, which has been one of the challenging tasks for implementation of FD in radiofrequency communication. On the other hand, the diffraction of VLC is sufficiently small that the effect of self interference is ignored. Therefore, as mentioned in the manuscript, it is easier to employ FD wireless communications in VLC networks than in radio communication networks.
In the revised manuscript, we added the explanations in Section 2.2.
(1-4)
In section 6. Conclusion the authors explain the improvements of this paper but, in my opinion, the authors could report that the system works correctly for a reduced number of devices connected simultaneously (N = 5) if, for example, this number increases (N = 10), the percentage of discards is multiplied by 5 even reducing the value of K.
Answer for (1-4)
Please refer to the table in the PDF of the answer.
Table 1 shows the revised table, which newly includes the packet discard probability of the conventional scheme (K = 0). In the proposed scheme, the access delay increases as the number of devices increases. Therefore, we need to reduce the limitation for keeping BE and NB K to satisfy the required access delay. As a result, the packet discard probability for N = 5 is multiplied by about 5 when N = 10. However, from Table 1, more than 30 % of packets are discarded even in the non-saturated condition (15 Mbps) in the conventional scheme (K = 0). It is seen from Table 1 that the proposed scheme significantly improves the packet discard probability compared with the conventional scheme. Therefore, even when the number of devices is large, the proposed scheme can achieve the required access delay, low packet discard probability, and high network throughput.
In the revised manuscript, we added the Table 1 in this answer letter to the revised manuscript (Table 2 in Section 5.2), and we added the above explanations in Section 5.

Reviewer 2 Report
The authors presented an improved backoff scheme and performance evaluations in terms of full-duplex MAC protocol in VLC systems. For performance evaluations, the authors provided a Markov-chain model and derived probability well.
However, the authors have not explained related works enough. The contribution of the paper should be revealed more by comparing to Ref [3,4,7,10,12, 18-22].
- Please put more explanation on Ref [3,4,7,10,12] to compare to the proposed scheme in section 2.1
- Please put more explanation on Ref [10,18-22] in section 4.2. And the authors need to put the explanation in section 2, related work, not only in section 4.2.
Author Response
(2-1)
Please put more explanation on Ref [3,4,7,10,12] to compare to the proposed scheme in section 2.1.
Answer for (2-1)
All references of [3,4,6,7,10,12] point out the effect of hidden devices on network performance. Especially, Refs. [3,4,12] investigate the detailed effects induced by hidden devices in VLC networks. On the other hand, Refs. [6,7,10] focus on
mitigating the hidden device problem by applying FD communication to VLC networks. The contents of each reference are as follows: Ref. [3] investigates the effect of hidden devices in uplink communication in VLC networks. Ref. [4] investigates the effect of the problem of the hidden device in IEEE 802.15.7 standard, where the authors simulate the uplink system performance using a slotted CSMA random access procedure. Ref. [12] evaluates performance in three situations: no hidden devices, a single hidden device, and no visibility between devices. Ref. [6] evaluates the performance for IEEE 802.15.7 based FD VLC networks under the use of a busytone signal. Ref. [7] proposes two protocols, U-ALOHA and FD-CSMA, to avoid the hidden device problem. This work avoids hidden device collisions by treating downlink communication as a busytone signal for uplink communication. However, data packet collisions remain due to the contentions among the data-packet transmissions simultaneously from multiple devices. For suppressing wasted time due to the collisions, exchange of control frames, RTS/CTS is applied to FD-CSMA protocols for VLC networks in [10]. Following the reviewer’s point out, we added the explanations of Refs. [3,4,12] to Section 2.1. Because Refs. [6,7,10] focus on mitigating the hidden device problem by applying FD communication to VLC networks, we added the explanations of Refs. [6,7,10] in Section 2.2. Note that only the statement related to Ref. [10] is colored by blue because the statement has already appeared in the original manuscript.
(2-2)
Please put more explanation on Ref [10,18-22] in section 4.2. And the authors need to put the explanation in section 2, related work, not only in section 4.2.
Answer for (2-2)
Refs. [10,18-22] research performance analysis for IEEE 802.15.7 networks. The contents of each reference are as follows: Ref. [21] proposed a priority-based MAC protocol for IEEE 802.15.7 networks and analyzed its performance using the traditional Markov-chain model proposed by Bianchi. Ref. [22] also uses the Markov-chain model to analyze the network performance under non-saturated conditions. However, the duration of packet transmission and backoff duration are not considered in the model of Refs. [21] and [22]. Ref. [18] has proposed the enhanced model to consider the detail operations in IEEE 802.15.7 MAC. This model enables us to consider the duration of packet transmission and backoff duration, which is one of the important factors for considering the behavior of IEEE 802.15.7 networks. The model of [18] assumes that the probability of sensing a channel busy by a device is independent of the backoff stage and the value of the backoff counter. This assumption may cause errors between analytical results and simulation ones. In order to enhance the accuracy of the analytical model, a semi-analytic approach is used in the model of Ref. [19]. The model of Ref. [20] considers more detail operations in IEEE 802.15.7 networks by extending the model of Ref. [19]. In addition, Ref. [20] analyzes the effect of using CCA twice when using a slotted CSMA/CA random access procedure defined in the IEEE 802.15.7 standard. However, Refs. [18-20] assume no hidden devices and networks under saturated conditions. Ref. [10] analyzed the performance of the FD-RTS/CTS scheme by extending the model of [18]. Ref. [10] assumes IEEE 802.15.7 networks with hidden devices. Following the reviewer’s point out, we added the above explanations in a subsection of “Performance analysis for IEEE 802.15.7 networks,” which is newly added as Section 2.3. At the References in the revised manuscript, additionally, we changed the order of those references by following those appearances. They are colored by blue.

Round 2
Reviewer 2 Report
It is well-improved and can be accepted.